# Effect of Inter-System Coupling on Heat Transport in a Microscopic Collision Model

**DOI:** 10.3390/e23040471

**Published:** 2021-04-16

**Authors:** Feng Tian, Jian Zou, Lei Li, Hai Li, Bin Shao

**Affiliations:** 1School of Physics, Beijing Institute of Technology, Beijing 100081, China; 3120195767@bit.edu.cn (F.T.); sbin610@bit.edu.cn (B.S.); 2School of Physical Science and Technology, Inner Mongolia University, Hohhot 010021, China; lilei@imu.edu.cn; 3School of Information and Electronic Engineering, Shandong Technology and Business University, Yantai 264005, China; lihai@sdtbu.edu.cn

**Keywords:** inter-system coupling, quantum correlation, heat current, thermal rectification, collision model

## Abstract

In this paper we consider a bipartite system composed of two subsystems each coupled to its own thermal environment. Based on a collision model, we mainly study whether the approximation (i.e., the inter-system coupling is ignored when modeling the system–environment interaction) is valid or not. We also address the problem of heat transport unitedly for both excitation-conserving system–environment interactions and non-excitation-conserving system–environment interactions. For the former interaction, as the inter-system interaction strength increases, at first this approximation gets worse as expected, but then counter-intuitively gets better even for a stronger inter-system coupling. For the latter interaction with asymmetry, this approximation gets progressively worse. In this case we realize a perfect thermal rectification, and we cannot find an apparent rectification effect for the former interaction. Finally and more importantly, our results show that whether this approximation is valid or not is closely related to the quantum correlations between the subsystems, i.e., the weaker the quantum correlations, the more justified the approximation and vice versa.

## 1. Introduction

In most practical situations, a quantum system inevitably interacts with its environment, which induces decoherence and dissipation [1]. In this case, its dynamics are usually described by the Gorini–Kossakowski–Lindblad–Sudarshan (GKLS) quantum master equation with a series of approximations [2]. When deriving a master equation for an open quantum system, one may always obtain a global master equation, which considers the full system Hamiltonian, i.e., including the direct coupling between the subsystems [3,4,5,6]. However, such derivation is very complicated when the system is composed of two or more interacting subsystems. Hence a local master equation that ignores the direct interactions between the subsystems is often used as a substitute [3,4,5,6,7,8,9,10,11]. By means of these two kinds of master equations, a lot of efforts has recently been devoted to the heat transport for thermodynamic systems [7,8,9,10,11,12,13,14,15]. However, by comparing the dynamics resulting from the corresponding master equations with exact numerical simulations, both approaches may lead to seeming thermodynamic inconsistencies or just suit to some parameter regimes, as pointed out in References [15,16,17,18,19,20,21,22]. In Reference [3], the local description for two coupled quantum nodes may predict heat currents from a cold to a hot thermal reservoir, or the existence of currents even in the absence of a temperature gradient. The origin of these effects, as discussed in References [5,23], lies in the fact that there is an external work cost related to the breaking of global detailed balance. By including this work cost, this inconsistencies can be resolved. For weak inter-system coupling, it was shown that the global approach fails in non-equilibrium situations, whereas the local approach agrees with the exact solution [19]. In Reference [21], due to a failure of the secular approximation, it was reported that the global master equation erroneously gave a vanishing heat current through a spin-$\frac{1}{2}$ Heisenberg chain in the presence of a finite-temperature gradient. For two coupled qubits interacting with common and separate baths, it was shown that the global approach with partial secular rather than full secular approximation always provides the most accurate choice for the master equation [24]. In Reference [25], it was shown that the completely positive version of the Redfield equation and an appropriate time-dependent convex mixture of the local and global solutions gives rise to the most accurate semigroup approximations of the exact system dynamics. Recently, it was proved that local master equations are consistent with thermodynamics without resorting to a microscopic model [26]. Moreover, the two approaches above, from the viewpoint of the system, often rely on approximate Markovian master equations derived under the assumptions of weak system–environment coupling, which would become challenging under strong coupling.

Furthermore, the manipulation of heat transport in non-equilibrium steady-state has been identified as one of the crucial studies of quantum thermodynamics, which gives us an improved understanding of classical thermodynamics in quantum domain [27,28,29,30,31,32,33,34,35]. For example, heat transport between two bosonic reservoirs was predicted for a coupled two-state system, and a formula for thermal conductance was derived based on a rate equation formalism [33]. Bandyopadhyay et al. proposed a numerical scheme for exactly simulating the heat transport in a quantum harmonic chain with self-consistent reservoirs [34]. By means of an effective harmonic Hamiltonian, a quantum thermal transport through anharmonic systems was studied within the framework of the nonequilibrium Green’s function method [35]. Besides, quantum devices such as heat rectifier, thermal memory, and thermal ratchet, have also become goals of controlling thermal transport in quantum thermodynamics [36,37,38,39,40,41,42,43,44,45]. It was found that, by using the quantum master equation, thermal rectification in anisotropic Heisenberg spin chains could change sign when the external homogeneous magnetic field was varied [43]. An optimal rectification in the ultrastrong-coupling regime of two coupled two-level systems was shown [44]. In Reference [45], Jose et al. studied two interacting spin-like systems characterized by different excitation frequencies, which can be used as a quantum thermal diode.

Recently, the collision model, also called repeated interactions, has drawn attention for its potential advantage in simulating open quantum system [46,47,48,49,50,51,52,53,54,55,56,57]. It was assumed that the environment consists of a large collection ancillae and the system of interest interacts, or collides, with an ancilla at each time step. In the framework of collision model, a continuous-time description in terms of a Lindblad master equation can be derived in the short-time limit provided some assumptions are made about the system–ancilla interaction [40,49]. For instance, in Reference [58] the system’s dynamics embodied by the stroboscopic map can be approximated by a Lindblad master equation in a short-time limit. In a similar way, in Reference [5] a local master equations with a Lindblad form for two coupled harmonic oscillator was derived by using the method of repeated interactions. Moreover, also from the viewpoint of the system, the corresponding reduced dynamics can be obtained in many cases without any approximations [59,60,61,62,63,64,65]. This is because collision model allows for the possibility to decompose a complicated open dynamics in terms of discrete elementary processes. It is particularly suited for addressing the thermodynamics of engineered reservoirs [23,48,49,62,66,67]. For example, under an energy-preserving system–environment interaction within the framework of collision model, it has been found that the non-monotonic time behavior of the heat exchange between the system and environment could serve as an indicator of non-Markovian behaviors [62]. Besides, based on a microscopic collision model, heat current between a coupled system can flow from the cold nonthermal reservoir to the hot one due to the contribution of coherence [66]. A link between information and thermodynamics, for a multipartite open quantum system with a finite temperature reservoir, was displayed in term of a collision model [67].

In this paper, we consider a two coupled qubit system that interacts with its local environment consisting of a large collection of identical ancillae. Inspired by the previous work on the local and global master equations, we consider an decoupling approximation within the framework of collision model, specifically, environment acts on subsystem without considering the inter-system coupling, i.e., ignores the direct coupling between the subsystems when modeling the system–environment interaction. It is noted that although this approximation in this collision model is not strictly a substitute for the local master equation we described above, we expect that the results derived from this simple and solvable model can provide a reference. We mainly study whether this decoupling approximation is valid or not, i.e., whether or when this direct coupling can be ignored. We systematically examine the heat transport through the system in both weak and strong system–ancilla coupling. Within the framework of collision model, we give a more general definition of heat current under non-energy preserving system–ancilla interactions (there is external work performing on the whole system–ancilla compound). In the case of excitation-conserving system–ancilla interactions, we find that the results predicted with the decoupling approximation is valid even at strong inter-system interactions. In particular, we realize a perfect rectification in the asymmetric systems and give a discussion about the mechanism of this phenomenon. Moreover, we find that whether or not this decoupling approximation is valid is closely related to the quantum correlations between the subsystems.

## 2. Model and Methods

We consider a bipartite system *S* consisting of two identical two-level subsystems $S_{i}$(i=1,2), and each subsystem is coupled to its local thermal subenvironment $E_{i}$ set at temperature $T_{i}$. Here the subenvironment $E_{i}$ is a sequence of non-interacting ancillae $\left(E_{1}^{i},E_{2}^{i},...,E_{n}^{i}\right)$ all in the same initial state $\eta_{n}^{i}$. Subsystem $S_{i}$ interacts with the connected subenvironment $E_{i}$ via a series of short subsystem–ancilla interactions. The joint state of system and environment is initially factorized:(1)$$\rho^{SE}\left(0\right)=\rho^{S}\left(0\right)\left({⊗}_{j=1}^{n}\eta_{j}^{1}\right)\left({⊗}_{j=1}^{n}\eta_{j}^{2}\right)$$
where $\rho^{S}\left(0\right)$ and ${⊗}_{j=1}^{n}\eta_{j}^{i}$ are the initial states of the open system and subenvironment $E_{i}$, respectively.

In Figure 1 we show a schematic sketch of the collision model considered. It is composed of a series of repeated rounds. In any one round of the collision process, first the whole system *S* undergoes a free evolution lasting a time interval τ, subsequently $S_{1}$ and $S_{2}$ locally interact with only one ancilla of its environment, respectively. Then by tracing out the environment’s degrees of freedom and repeating the above process in the next round, we can obtain the reduced state of the system in the full time evolution. Therefore, each round is composed of three steps: a free evolution of *S* and two subsystem–ancilla interactions. It is noted that the subenvironment is assumed to be large enough, so that the subsystem never collides twice with the same ancilla. As a consequence, at each collision round *n*, the subsystem $S_{i}$ collides with a ”fresh” $E_{n}^{i}$.

We assume throughout this paper that *S* and each ancilla $E_{n}^{i}$ of the subenvironment $E_{i}$ are qubits with logical states {|0〉,|1〉}. The corresponding free Hamiltonians for the subsystem $S_{i}$(i=1,2) and the ancilla $E_{n}^{i}$ are ${\hat{H}}_{S_{i}}=\frac{1}{2}\omega_{i}{\hat{\sigma }}_{z}$ and ${\hat{H}}_{E_{n}^{i}}=\frac{1}{2}\omega_{0}{\hat{\sigma }}_{z}$, respectively. Here ${\hat{\sigma }}_{z}$ is the usual Pauli operator (we set ℏ=1). The free dynamics of the whole system *S* are described by the unitary evolution operator
(2)$${\hat{U}}_{S_{1},S_{2}}=\exp\left[-i\left({\hat{H}}_{0}+{\hat{H}}_{int}^{S_{1},S_{2}}\right)\tau \right],$$
with ${\hat{H}}_{0}={\hat{H}}_{S_{1}}+{\hat{H}}_{S_{2}}$ and the inter-system interaction Hamiltonian ${\hat{H}}_{int}^{S_{1},S_{2}}$. We use unitary operator ${\hat{V}}_{S_{i},E_{n}^{i}}$ to model the collisions between the subsystem $S_{i}$ and ancilla $E_{n}^{i}$ of subenvironment $E_{i}$ (Its exact definition will be given later in different cases); it is assumed that all the subsystem–ancilla collisions have the same duration τ. So the dynamical map ${\hat{\Lambda }}_{S_{1},S_{2}}$ that governs the free evolution of the system *S* and map ${\hat{\Psi }}_{S_{1},E_{n}^{i}}$ that governs the system–subenvironment interaction at the *n*th round can be written as
(3)$${\hat{\Lambda }}_{S_{1},S_{2}}\left(\rho \right)={\hat{U}}_{S_{1},S_{2}}\rho {\hat{U}}_{S_{1},S_{2}}^{†},$$
(4)$${\hat{\Psi }}_{S_{i},E_{n}^{i}}\left(\rho \right)={\hat{V}}_{S_{i},E_{n}^{i}}\rho {\hat{V}}_{S_{i},E_{n}^{i}}^{†},$$
respectively. Following the repeated interaction approach mentioned above, the joint state of system and ancillae is brought from the *n*th round to the (n+1)th round through the process
(5)$$\rho_{n-1}^{S}⊗\eta_{n}^{1}⊗\eta_{n}^{2}\to \rho_{n}^{SE}=\hat{U}\left[\rho_{n-1}^{S}⊗\eta_{n}^{1}⊗\eta_{n}^{2}\right]{\hat{U}}^{†},$$
where $\hat{U}={\hat{V}}_{S_{2},E_{n}^{2}}{\hat{V}}_{S_{1},E_{n}^{1}}{\hat{U}}_{S_{1},S_{2}}$. Hence after round *n*, the reduced density matrix of the system $\rho_{n}^{S}$ is
(6)$$\rho_{n}^{S}={\mathrm{Tr}}_{E_{n}^{1},E_{n}^{2}}\left[\rho_{n}^{SE}\right],$$
where ${\mathrm{Tr}}_{E_{n}^{1},E_{n}^{2}}\left[\cdot \right]$ denotes the partial trace over the two ancillae $E_{n}^{1}$ and $E_{n}^{2}$. Similarly, the reduced state ${\tilde{\eta }}_{n}^{1(2)}$ of the *n*th ancilla of subenvironment $E_{1(2)}$ is
(7)$${\tilde{\eta }}_{n}^{1(2)}={\mathrm{Tr}}_{SE_{n}^{2(1)}}\left[\rho_{n}^{SE}\right].$$

Throughout we assume each ancilla of subenvironment $E_{i}$ to be initially in a thermal state with inverse temperature $\beta_{i}=1/T_{i}$ (we set k=1), namely,
(8)$$\eta_{n}^{i}=\frac{1}{Z}\exp\left(-\beta_{i}{\hat{H}}_{E_{n}^{i}}\right),$$
where $Z=\mathrm{Tr}[\exp\left(-\beta_{i}{\hat{H}}_{E_{n}^{i}}\right)]$ is the partition function.

In the *n*th round of the dynamics, the whole system *S* undergoes a free evolution; $S_{1}$ interacts with $E_{n}^{1}$; next, $S_{2}$ interacts with $E_{n}^{2}$; then $S_{1}$ and $S_{2}$ shift by one site to the (n+1)th round in which also after the free evolution of system $S_{1}$-$E_{n+1}^{1}$ and $S_{2}$-$E_{n+1}^{2}$ interactions subsequently take place. This process will repeat to the (n+2)th round. The free evolution is governed by Equations (Equation 2), and now we begin to give the subsystem–ancilla evolution operator. When ancilla $E_{n}^{i}$ collides with subsystem $S_{i}$ not ignoring the direct interaction between the two subsystems, the corresponding unitary time evolution operator can be written as
(9)$${\hat{V}}_{S_{i},E_{n}^{i}}=\exp\left[-i\left({\hat{H}}_{S_{1}}+{\hat{H}}_{int}^{S_{1},S_{2}}+{\hat{H}}_{S_{2}}+{\hat{H}}_{E_{n}^{i}}+{\hat{H}}_{int}^{S_{i},E_{n}^{i}}\right)\tau \right],$$
where ${\hat{H}}_{int}^{S_{i},E_{n}^{i}}$ is the interaction Hamiltonian between the subsystem $S_{i}$ and ancilla $E_{n}^{i}$. However, it is convenient to ignore the direct interaction between the two subsystems, namely, a decoupling approximation neglects the coupling between the subsystems when modeling the system–environment interaction. In this case, the corresponding unitary time evolution operator is written as
(10)$${\hat{V}}_{S_{i},E_{n}^{i}}^{app}=\exp\left[-i\left({\hat{H}}_{S_{i}}+{\hat{H}}_{E_{n}^{i}}+{\hat{H}}_{int}^{S_{i},E_{n}^{i}}\right)\tau \right].$$

It is obvious that the difference between Equations (Equation 9) and (Equation 10) resides in the inter-system interaction in each round: Equation (Equation 9) arises naturally when modeling the subenvironment–subsystem coupling from a microscopic model considering the full system Hamiltonian, i.e., not ignore the direct interaction between its subsystems, while Equation (Equation 10) ignores this interaction, i.e., the decoupling approximation.

## 3. Symmetric System

In this section we consider a symmetric system, from which to investigate whether the inter-system interaction can be ignored or not. Among the possible choices for the interaction between the subsystem $S_{i}$ and their connected ancilla $E_{n}^{i}$, we choose the excitation-conserving interaction Hamiltonian as
(11)$${\hat{H}}_{int}^{S_{i},E_{n}^{i}}=\gamma \left({\hat{\sigma }}_{x}⊗{\hat{\sigma }}_{x}+{\hat{\sigma }}_{y}⊗{\hat{\sigma }}_{y}\right)$$
with subsystem–subenvironment interaction strength γ. The interaction between $S_{1}$ and $S_{2}$ takes the same form
(12)$${\hat{H}}_{int}^{S_{1},S_{2}}=\delta \left({\hat{\sigma }}_{x}⊗{\hat{\sigma }}_{x}+{\hat{\sigma }}_{y}⊗{\hat{\sigma }}_{y}\right)$$
with coupling strength δ. We consider that the energy gaps of subsystems are the same, i.e., $\omega_{1}=\omega_{2}=\omega_{0}$.

### 3.1. Heat Current

Now we investigate the heat currents to evaluate the performance of the decoupling approximation. We also consider the heat currents in the case of not ignoring the inter-system interactions (i.e., without the decoupling approximation), which serves as a benchmark. In the case of decoupling approximation, for the system characterized by Hamiltonians of Equations (Equation 11) and (Equation 12), its corresponding unitary system–environment operator preserves the energy, i.e.,
(13)$$[{\hat{V}}_{S_{i},E_{n}^{i}}^{ign},{\hat{H}}_{S_{i}}+{\hat{H}}_{E_{i}}]=0.$$

This implies that all the energy leaving the ancilla enters the system. Hence, during the (n+1)th round, the heat exchange ▵Q between system $S_{1(2)}$ and ancilla $E_{n}^{1(2)}$ is given by
(14)$$▵Q_{E_{n}^{1(2)}}=\mathrm{Tr}\left[{\hat{H}}_{E_{n}^{1(2)}}\left({\tilde{\eta }}_{n}^{1(2)}-\eta_{n}^{1(2)}\right)\right].$$

Due to $▵Q_{E_{n}^{1}}=-▵Q_{E_{n}^{2}}$ for steady state, the stationary heat current flowing from subenvironment $E_{1}$ to the system $S_{1}$ can be defined as
(15)$$J_{h}=-▵Q_{E_{n}^{1}}.$$

This definition was often used when Equation (Equation 13) is satisfied [42,43,44,45,58,66]. However, in the case of not ignoring the inter-system interactions, the corresponding unitary system–environment operator no longer preserves the energy because of
(16)$$[{\hat{V}}_{S_{i},E_{n}^{i}}^{con},{\hat{H}}_{S_{1}}+{\hat{H}}_{S_{2}}+{\hat{H}}_{int}^{S_{1},S_{2}}+{\hat{H}}_{E_{i}}]\neq 0.$$

That is, not all the energy leaving the ancilla enters the system. So an external work is required to turn the system–ancilla coupling on and off [67,68], which we refer to as the switching work (labeled $W_{SW}$). In each round of collisions illustrated in Figure 1, the corresponding work $W_{SW}$ on the system can be written as
(17)$$W_{SW}=W_{1}+W_{2}.$$

The first term $W_{1}$ is from the sudden on and off switching of $S_{1}-E_{n}^{1}$ interaction, and it reads
(18)$$W_{1}=\mathrm{Tr}\left[{\hat{H}}_{int}^{S_{1},E_{n}^{1}}\left(\rho_{S_{1}S_{2}}^{\prime }⊗\eta_{n}^{1}-\rho_{S_{1}S_{2}E_{n}^{1}}\right)\right],$$
where $\rho_{S_{1}S_{2}}^{\prime }$ is the state of system before $S_{1}-E_{n}^{1}$ collision, and $\rho_{S_{1}S_{2}E_{n}^{1}}$ is the global state of system *S* and ancilla $E_{n}^{1}$ after $S_{2}-E_{n}^{1}$ collision. Similarly, the second term associated with $S_{2}-E_{n}^{2}$ interaction takes the form
(19)$$W_{2}=\mathrm{Tr}\left[{\hat{H}}_{int}^{S_{2},E_{n}^{2}}\left(\rho_{S_{1}S_{2}}^{\prime \prime }⊗\eta_{n}^{2}-\rho_{S_{1}S_{2}E_{n}^{2}}\right)\right],$$
where $\rho_{S_{1}S_{2}}^{\prime \prime }$ is the reduced state of system after $S_{1}-E_{n}^{1}$ collision, and $\rho_{S_{1}S_{2}E_{n}^{2}}$ is the global state of system *S* and ancilla $E_{n}^{2}$ after $S_{2}-E_{n}^{2}$ collision. As a result, Equation (Equation 15) is no longer valid to calculate the heat current in the case of not ignoring the inter-system interactions.

Based on this collision model, we focus on deriving the expression of heat current that can be applicable in the cases of ignoring and not ignoring the inter-system interactions, as well as weak and strong system–environment couplings. Physically, heat current is defined as the energy going through the system. We consider the energy change of the system in a complete round (i.e., free evolution of *S*, $S_{1}-E_{n}^{1}$ and $S_{2}-E_{n}^{2}$ interactions). The energy change $▵E_{S}^{1}$ of system *S* in each $S_{1}-E_{n}^{1}$ interaction can be written as
(20)$$▵E_{S}^{1}=\mathrm{Tr}\left[\left({\hat{H}}_{0}+{\hat{H}}_{int}^{S_{1}S_{2}}\right)\left(\rho_{S_{1}S_{2}}^{\prime \prime }-\rho_{S_{1}S_{2}}^{\prime }\right)\right].$$

Similar expressions can be obtained for $S_{2}-E_{n}^{2}$ interaction, and the corresponding energy change $▵E_{S}^{2}$ of system *S* is given by
(21)$$▵E_{S}^{2}=\mathrm{Tr}\left[\left({\hat{H}}_{0}+{\hat{H}}_{int}^{S_{1}S_{2}}\right)\left(\rho_{S_{1}S_{2}}^{\prime \prime \prime }-\rho_{S_{1}S_{2}}^{\prime \prime }\right)\right],$$
where $\rho_{S_{1}S_{2}}^{\prime \prime \prime }$ is the reduced state of system *S* after $S_{2}-E_{n}^{2}$ collision. Since the inter-system dynamics is unitary, the energy of system is preserved in each free evolution. For steady state, the state of the system cannot change in a complete round, leading to
(22)$$▵E_{S}^{1}=-▵E_{S}^{2}.$$

From the above discussion, it is clear that $▵E_{S}^{1}$ is the energy going through the system. Thus, the stationary heat current $J_{h}$ flowing from $E_{1}$ to $E_{2}$ can be rewritten as
(23)$$J_{h}=▵E_{S}^{1}.$$

In the case of ignoring the inter-system interactions, if the corresponding system satisfies the commutation condition Equation (Equation 13), i.e., all energy changes in the system can be attributed to energy flowing to or from the ancilla, it is thus clear that Equations (Equation 23) reduces to Equation (Equation 15).

Here we should mention that the exchange of energy between two quantum systems, characterized by particular commutation relation between the local Hamiltonians and the interaction operator, can always be split into work and heat [69,70]. This formalism was extended to situations where one, or both, subsystems are coupled to a thermal environment. In the case of not ignoring the inter-system interaction, according to Reference [70] the stationary heat current and work in each $S_{1}-E_{n}^{1}$ collision are calculated as
(24)$$J_{h}^{\prime }=-i\int_{0}^{\tau }\mathrm{Tr}\left[I⊗\left({\hat{H}}_{0}+{\hat{H}}_{int}^{S_{1},S_{2}}\right),{\hat{H}}_{int}^{S_{1},E_{n}^{1}}⊗I\right]C_{12}dt,$$
(25)$$W=-i\int_{0}^{\tau }\mathrm{Tr}\left[{\hat{H}}_{S}^{eff},{\hat{H}}_{S}\right]{\tilde{\eta }}_{n}^{1}dt,$$
where $C_{12}$ represents the correlations between the system *S* and $E_{n}^{1}$ after $S_{1}-E_{n}^{1}$ collision, and ${\hat{H}}_{S}^{eff}$ is the diagonal part of matrix ${\mathrm{Tr}}_{E_{n}^{1}}\left[{\hat{H}}_{int}^{S_{1},E_{n}^{1}}\left({\tilde{\eta }}_{n}^{1}⊗I\right)\right]$. From numerical calculations, we check that the heat current $J_{h}^{\prime }$ given by Equation (Equation 24) is equal to that given by Equation (Equation 23). Thus our previous definition of Equation (Equation 23) is consistent with Reference [70].

In Figure 2 we plot $J_{h}$ as a function of the inter-system coupling strength δ for various subsystem–ancilla coupling strength γ in the cases with and without the decoupling approximation. We fix the initial state of the system to be |11〉 and set $T_{1}=5\omega_{0}$ and $T_{2}=\omega_{0}$ for subenvironments $E_{1}$ and $E_{2}$, respectively. It can be seen from Figure 2 that, for fixed γ, the heat currents in both cases increase rapidly with the increase of δ and they eventually reach the same steady value, before which the heat current from the decoupling approximation is always smaller than that without the approximation. In general, for weak inter-system coupling, one would expect smaller deviation from the case without the approximation. As expected, in the limit δ→0, it gives the same (zero) heat current as that without the approximation. For fixed γ, when δ increases we observe that the heat current predicted with the decoupling approximation gradually deviates from that without the approximation, and this deviation quickly increases from zero to its maximum. However, it might seem counter-intuitive that this deviation gradually decreases with δ and eventually vanishes at larger values of δ. That is, for larger δ, the heat currents from the decoupling approximation can still be consistent with that without the approximation, and this decoupling approximation consequently may not necessarily break down.

Moreover, we find that the difference of the heat currents between those with and without the decoupling approximation strongly depends on γ, as shown in Figure 2. A bigger difference is obtained for stronger coupling strength γ. For instance, for $\gamma =0.8\omega_{0}$, the difference that maintains nonzero values is within a larger region of δ, compared with $\gamma =0.2\omega_{0}$ and $0.5\omega_{0}$. That is, when assessing the heat currents the results from the decoupling approximation is inconsistent with that without this approximation. Smaller difference is obtained for weaker coupling strength γ. For $\gamma =0.2\omega_{0}$, it can be seen from the inset of Figure 2 that the difference between two cases maintaining nonzero values is within a very small region below $\delta ∼0.04\omega_{0}$. In other words, for smaller values of γ, the results from the decoupling approximation agrees well with that without this approximation, and consequently it also approximately predicts the correct heat currents within almost all region of δ. Physically, this can be easily understood as following: decreasing the system–ancilla coupling strength would weaken the influence of the inter-system coupling ignored.

### 3.2. Trace Distance

When investigating the accuracy of the local and global master equations, steady state is often used as a reference for predicting the results of the nonequilibrium dynamics [16,19]. To further assess the validity of decoupling approximation, we also consider the obtained steady states. As a measure of distinguish ability, here we examine the trace distance between the steady state obtained from the case with the decoupling approximation and the steady state obtained without this decoupling approximation:(26)$$D_{T}=\frac{1}{2}{\left\|\rho^{S}-\rho^{Sapp}\right\|}_{1},$$
where ${\left\|\cdot \right\|}_{1}$ is the trace norm while $\rho^{S}$ and $\rho^{Sapp}$ are the reduced steady states of the system with and without the decoupling approximation, respectively. The trace distance is equal to unity for fully distinguishable states, while it is null for identical states. Figure 3 shows the dependence of $D_{T}$ on δ for different strength γ. For fixed γ, it can be seen that $D_{T}$ quickly increases and then gradually decreases with the increase of δ (e.g., for $\gamma =0.1\omega_{0}$). That is, when δ increases, the steady state $\rho_{n}^{Sapp}$ predicted with the decoupling approximation gradually deviates from the steady state $\rho_{n}^{S}$ predicted without this approximation, then this deviation reaches its maximum and eventually decreases. It is obvious that this behavior of the steady state from this approximation is indeed similar to that of heat current. Here we also arrive at a conclusion similar to that of heat current: even for a large δ, decoupling approximation can be well justified when predicting the steady state. The inset of Figure 3 plots the value of $\delta^{\prime }$ for the maximum of $D_{T}$ as a function of γ (i.e., when $\delta =\delta^{\prime }$, $D_{T}$ reaches its maximum for fixed γ). It can be seen that $\delta^{\prime }$ nonlinearly increases with the subenvironment–ancilla interaction strength γ.

Figure 3 also shows the effect of system–ancilla coupling strength γ on $D_{T}$. For $\gamma =0.2\omega_{0}$, it can be seen that $D_{T}$ maintains a larger value only within a smaller region of δ, and quickly decreases compared to $\gamma =0.5\omega_{0}$ and $\gamma =0.8\omega_{0}$. Again, for weak γ, steady state predicted with the decoupling approximation agrees well with that without the approximation even at strong interaction strengths δ. In other words, this approximation is well justified to describe steady state in this case.

### 3.3. Quantum Discord

Why can the decoupling approximation give a good estimate of heat current and steady state even for a large δ ? In References [71,72], it was found that the compositeness (two particles behave like a single particle) is closely related to the quantum correlations between the constituent particles. It is thus very interesting to consider the quantum correlations between the bipartite system $S_{1}$ and $S_{2}$, which can be captured by the discord [73]
(27)$$D\left(\rho_{S}\right)=\min_{\Pi_{A}}\left\{I\left(\rho_{S}\right)-I\left(\Pi_{A}\rho_{S}\right)\right\}.$$

Here $\Pi_{A}$ is a set of rank-one POVM projectors on system $S_{1}$, and $I\left(\rho_{S}\right)=S\left(\rho_{S}\right)-S\left(\rho_{S_{1}}\right)-S\left(\rho_{S_{2}}\right)$ is the quantum mutual information associated with the von Neumann entropy. Without the decoupling approximation, in Figure 4 we plot D as a function of δ. As inter-system coupling strength δ increases it can be seen that, for fixed γ, D first increases from zero to its maximum, then it gradually decreases, i.e., the quantum correlations between two subsystems first increases and then decreases. It is obvious that such behavior is associated with the heat current or the steady state, i.e., the greater the quantum correlation D between two subsystems, the greater the deviation of the heat current and steady state predicted with the decoupling approximation, and vice versa. In general, one would think that whether this approximation is valid or not should depend on inter-system coupling strength. In fact, inter-system interactions are really necessary and they serve as physical means to create quantum correlations, while it is noted that such correlations can still be present in spatially separated subsystem that no longer interact after it is created. When the inter-system coupling produces a stronger correlations, these two subsystems behave more like a composite and can not be treated separately, so this decoupling approximation gets worse. The fact that decoupling approximation can give an good description even at stronger inter-system coupling is because the stronger inter-system interaction does not produce a strong enough quantum correlations, as shown in Figure 4. To summarize, this approximation depends strongly on quantum correlations between the two qubits, i.e., the higher the quantum correlations, the worse the approximation, and vice versa.

Moreover, for this model, we also consider the other situations by replacing Equations (Equation 11) and (Equation 12) with any of the following three combinations of excitation-conserving interactions:${\hat{H}}_{int}^{S_{1},S_{2}}=\delta \left({\hat{\sigma }}_{x}⊗{\hat{\sigma }}_{x}+{\hat{\sigma }}_{y}⊗{\hat{\sigma }}_{y}+{\hat{\sigma }}_{z}⊗{\hat{\sigma }}_{z}\right)$ and ${\hat{H}}_{int}^{S_{i},E_{n}^{i}}=\gamma \left({\hat{\sigma }}_{x}⊗{\hat{\sigma }}_{x}+{\hat{\sigma }}_{y}⊗{\hat{\sigma }}_{y}\right)$.${\hat{H}}_{int}^{S_{1},S_{2}}=\delta \left({\hat{\sigma }}_{x}⊗{\hat{\sigma }}_{x}+{\hat{\sigma }}_{y}⊗{\hat{\sigma }}_{y}\right)$ and ${\hat{H}}_{int}^{S_{i},E_{n}^{i}}=\gamma \left({\hat{\sigma }}_{x}⊗{\hat{\sigma }}_{x}+{\hat{\sigma }}_{y}⊗{\hat{\sigma }}_{y}+{\hat{\sigma }}_{z}⊗{\hat{\sigma }}_{z}\right)$.${\hat{H}}_{int}^{S_{1},S_{2}}=\delta \left({\hat{\sigma }}_{x}⊗{\hat{\sigma }}_{x}+{\hat{\sigma }}_{y}⊗{\hat{\sigma }}_{y}+{\hat{\sigma }}_{z}⊗{\hat{\sigma }}_{z}\right)$ and ${\hat{H}}_{int}^{S_{i},E_{n}^{i}}=\gamma \left({\hat{\sigma }}_{x}⊗{\hat{\sigma }}_{x}+{\hat{\sigma }}_{y}⊗{\hat{\sigma }}_{y}+{\hat{\sigma }}_{z}⊗{\hat{\sigma }}_{z}\right)$.

For these three combinations, we also investigate the heat current, trace distance, and quantum discord with and without ignoring the inter-system interactions. The corresponding results are similar to those from Equations (Equation 11) and (Equation 12). It is noted that, with the decoupling approximation, the corresponding unitary operators of all three combinations above, like Equations (Equation 11) and (Equation 12), also satisfy the condition $[{\hat{V}}_{S_{i},E_{n}^{i}}^{ign},{\hat{H}}_{S_{i}}+{\hat{H}}_{E_{i}}]=0$, i.e., they correspond to energy-preserving system–environment interactions and no external work exists. Without the approximation, in each round (i.e., free evolution of *S*, $S_{1}-E_{n}^{1}$ and $S_{2}-E_{n}^{2}$ interactions) although there is an external work due to $[{\hat{V}}_{S_{i},E_{n}^{i}}^{con},{\hat{H}}_{S_{1}}+{\hat{H}}_{S_{2}}+{\hat{H}}_{int}^{S_{1},S_{2}}+{\hat{H}}_{E_{i}}]\neq 0$, after some calculations we find that this external work in each round is much smaller than the heat flowing through the system in each round.

## 4. Asymmetric System

In this section, we turn our attention to the systems featured by distinct asymmetries, such as the qubits with different frequencies or asymmetric couplings to their reservoirs. Indeed, inspired by the manipulation and control of the thermal transport in micro-scale, such asymmetric systems were widely investigated in the thermal diode and thermal transistor [37,38,39,40,41,42,43,44,45]. In the following, we will study the heat transport and, at the same time, study whether the inter-system interaction can be ignored or not when modeling the system–environment interaction.

### 4.1. Off-Resonant Interacting Qubits

First we consider two qubits are characterized by different energy gaps $\omega_{1}$ and $\omega_{2}$, i.e., $\omega_{1}\neq \omega_{2}$. The interaction Hamiltonian between two subsystems $S_{1}$ and $S_{2}$ is given as
(28)$${\hat{H}}_{int}^{S_{1},S_{2}}=\delta {\hat{\sigma }}_{z}⊗{\hat{\sigma }}_{z}.$$

Such system may arise, for example, when a nonuniform magnetic field is applied to a pair of interacting spins in the *z* directions. The coupling to each ancilla of environment is described by the following non-excitation-conserving Hamiltonian:(29)$${\hat{H}}_{int}^{S_{i},E_{n}^{i}}=\gamma {\hat{\sigma }}_{x}⊗{\hat{\sigma }}_{x}.$$

For this interaction, in both cases with and without the decoupling approximation, not all the energy leaving the environment enters the system due to $[{\hat{V}}_{S_{i},E_{n}^{i}}^{ign},{\hat{H}}_{S_{i}}+{\hat{H}}_{E_{i}}]\neq 0$ and $[{\hat{V}}_{S_{i},E_{n}^{i}}^{con},{\hat{H}}_{S_{1}}+{\hat{H}}_{S_{2}}+{\hat{H}}_{int}^{S_{1},S_{2}}+{\hat{H}}_{E_{i}}]\neq 0$.

From Equation (Equation 23) we calculate the stationary heat currents for system characterized by Hamiltonians of Equations (Equation 28) and (Equation 29), also addressing with and without the decoupling approximation separately. From our study we find that the heat current from the decoupling approximation is always zero even in the presence of a finite temperature gradient $T_{1}>T_{2}$ or $T_{2}>T_{1}$, regardless of the coupling strength δ and γ. This result deviates from that without the approximation, in which heat currents increases as δ increases, so the decoupling approximation gets progressively worse with the increase of δ.

Moreover, for this model we compare the heat currents for various temperature difference ▵T between two environments, which shows an asymmetric heat transport of the system (i.e., heat rectification). To quantify this rectification efficiency, we also give a rectification factor *R* defined as follows [43,44,45]:(30)$$R=\frac{|J_{h}\left(▵T\right)+J_{h}\left(-▵T\right)|}{\max(|J_{h}\left(▵T\right)|,|J_{h}\left(-▵T\right)|)},$$
where $J_{h}\left(▵T\right)$ is the forward heat current for $T_{1}>T_{2}$, and $J_{h}\left(-▵T\right)$ is the backward heat current when the temperature gradient is reversed. It can be seen from Figure 5 that in the case $T_{1}=10\omega_{0}$ and $T_{2}=8\omega_{0}$, its corresponding forward heat current is greater than the backward heat currents ($T_{2}=10\omega_{0}$ and $T_{1}=8\omega_{0}$). Hence an asymmetric conduction ($|J_{h}\left(▵T\right)|\neq |J_{h}\left(-▵T\right)|$) emerges, which varies with the increase of δ. If the temperature gradient raises, such as $T_{1}=10\omega_{0}$ and $T_{2}=4\omega_{0}$, this asymmetric conduction increases, i.e., the higher the gradient, the stronger the heat rectification. Especially, when $T_{1}=10\omega_{0}$ and $T_{2}=0.1\omega_{0}$, its corresponding backward heat current (cf. green dashed line corresponds to $T_{1}=0.1\omega_{0}$ and $T_{2}=10\omega_{0}$ in Figure 5) is almost zero for any δ, i.e., an optimal rectification is realized with R>0.996. Insets in Figure 5 plot factor *R* as a function of δ for various temperature differences ▵T. It can be seen that *R* increases with the increase of ▵T.

The above rectification phenomena can be explained as follows. There are two ingredients affecting the rectification effect. One is nonzero heat currents in the absence of temperature gradient and the other is the temperature difference ▵T between two environments. (i) It can be seen from Figure 6 that the case without the decoupling approximation can give $J_{h}>0$ even when the two environments are at the same temperature (cf. black dashed line corresponds to $T_{1}=T_{2}=10\omega_{0}$ in Figure 6). It seems to violate the second law of thermodynamics but is justified. This is because there is an external work which is closely related to the internal subsystem–ancilla interaction. By taking this into account, the thermodynamic consistency can be guaranteed. We also find that the heat transport through the two-qubit system is dramatically affected by this work cost and can even be inverted, leading to $J_{h}>0$ for $T_{1}<T_{2}$, i.e., heat current from a cold to a hot bath (cf. dashed lines in Figure 5). Similarly, in Reference [5] heat transport through a chain of harmonic oscillator is investigated by using a local master equation based on repeated interactions, and all thermodynamic inconsistencies (such as a heat current from the cold to the hot bath) can be resolved correctly with the consideration of external work turning the subsystem–ancilla coupling on and off. (ii) Because there is a non-zero heat current in the absence of temperature gradient (labeled $J_{0}$), so when $T_{1}>T_{2}$, it can be seen from Figure 6 that the corresponding heat current ($J_{h}>J_{0})$ increases with the increase of temperature gradient (cf. blue line in Figure 6). However, when $T_{1}<T_{2}$, as the temperature gradient increases, the corresponding heat current decreases from positive to zero (cf. red line in Figure 6). If the temperature difference increases further (if it is large enough), the corresponding heat current can change from positive to negative (cf. green line in Figure 6).

Figure 7 plots the trace distance $D_{T}$ between the density matrices with and without the decoupling approximation against δ for steady state. As expected, for δ=0, we observe that $D_{T}=0$, which means that this approximation gives the same steady state as that without the approximation. For δ≠0, $D_{T}$ increases with the increase of δ. Therefore, the steady state from the decoupling approximation would deviates from that without the approximation, and this deviation increases as δ increases.

Now we consider quantum discord of the bipartite system. Without the decoupling approximation, in Figure 8 we plot D, i.e., quantum correlations of two qubits, as a function of δ at steady state. It is clear that D increases with the increase of δ. This behavior can also be closely related to that of Figure 5 (or Figure 7), i.e., as δ increases, the heat currents (or the steady state) predicted with the decoupling approximation gradually deviates from that without the approximation. Unlike Section 3.3 where quantum correlations first increase and then decrease with the increase of inter-system coupling, it can be seen from Figure 8 that increasing inter-system coupling enforces stronger correlations, which means these two qubits behave more like a composite and must be treated as a whole, so the decoupling approximation gets worse with the increase of the inter-system interaction. This further confirms our conclusion in Section 3.3, i.e., the higher the quantum correlations between two qubits, the greater the deviation from not ignoring the inter-system interaction.

In addition, we also consider the other situations by replacing Equation (Equation 28) with ${\hat{H}}_{int}^{S_{1},S_{2}}=\delta \left({\hat{\sigma }}_{x}⊗{\hat{\sigma }}_{x}+{\hat{\sigma }}_{y}⊗{\hat{\sigma }}_{y}\right)$ or ${\hat{H}}_{int}^{S_{1},S_{2}}=\delta \left({\hat{\sigma }}_{x}⊗{\hat{\sigma }}_{x}+{\hat{\sigma }}_{y}⊗{\hat{\sigma }}_{y}+{\hat{\sigma }}_{z}⊗{\hat{\sigma }}_{z}\right)$. In spite of a quantitative difference, the corresponding results are similar to those obtained above.

In Section 3 we only consider a resonant two-qubit system (it is noted that the system–environment interactions we chose in Section 3 are different from Equation (Equation 29) in this section). For a more comprehensive study of heat transfer, now we consider two off-resonant qubits under the same interactions as in Section 3. From our numerical calculations we find that the corresponding results are similar to those obtained in Section 3, and are different from the results in this section above. That is, as δ increases, heat current predicted with the decoupling approximation gradually deviates from that without the approximation, then this deviation decreases at larger values of δ. Moreover, for the system–environment interactions in Section 3, we find that although there is also an asymmetric heat conduction in the case of without the approximation, no obvious thermal rectification emerges. This shows that thermal rectification are not only related to asymmetry of the system, but also related to the system–environment interactions.

### 4.2. Anisotropic Interacting Qubits

Connecting two qubits $S_{1}$ and $S_{2}$ by anisotropic interactions is another possible way to introduce asymmetry that may also exhibit rectification. Here we consider the case where the inter-system coupling is taken as an anisotropic exchange interaction [42]
(31)$${\hat{H}}_{int}^{S_{1},S_{2}}=\delta {\hat{\sigma }}_{z}⊗{\hat{\sigma }}_{x},$$
and the subsystem–ancilla coupling is chosen as Equation (Equation 29). From Equation (Equation 23), in Figure 9 we plot the heat current as a function of δ for these two cases. It can be seen that the heat current increases with the increase of δ. When $T_{1}>T_{2}$, the heat current obtained from the decoupling approximation (cf. red dashed line in Figure 9) gradually deviates from the corresponding one obtained without the approximation (cf. red solid line in Figure 9) with the increase of δ. We also investigate the corresponding trace distance $D_{T}$, and despite quantitative difference, its behaviors are similar to those of Figure 7, more specifically, as δ increases, the steady state predicted with the decoupling approximation gradually deviates from that without the approximation. So this approximation gets progressively worse as δ increases.

Moreover, in Figure 9, when $T_{1}<T_{2}$ we also consider the heat current without the decoupling approximation. It can be seen that in the case $T_{1}=10\omega_{0}$ and $T_{2}=\omega_{0}$ (cf. red solid line in Figure 9), its corresponding forward heat current is greater than the backward heat currents ($T_{2}=10\omega_{0}$ and $T_{1}=\omega_{0}$) (cf. blue solid line in Figure 9). So there is also a heat rectification, and the corresponding rectification factor is plotted in the inset of Figure 9, which decreases with the increase of δ. It can be seen that an optimal rectification is almost realized with a high rectification factor R>0.985. Besides, under the same excitation-conserving system–environment interactions as those of Section 3, we also compare the forward and backward heat currents for this asymmetric system. The corresponding result shows that, despite an asymmetric heat conduction, there is no obvious thermal rectification. From numerical calculations, we find that as long as the system–environment interactions are chosen to be like those in Section 3, no matter how we introduce asymmetry into the system, there is no apparent thermal rectification effect. Therefore, we conclude that the rectification effect strongly depends on the form of the system–environment interaction. It might be the reason why most work [42,43,44,45] chose system–bath interactions like Equation (Equation 29) to achieve the thermal rectification for various kinds of asymmetric systems.

Next we move to the quantum correlation D between two subsystems. Without the decoupling approximation, in Figure 10 we plot D as a function of δ at steady state. It is clear that its behavior can be closely related to the failure of the approximation. We again confirm the same conclusion: the weaker the correlation between the two subsystems, the more justified the decoupling approximation and vice versa.

## 5. Conclusions

In this paper we have considered a system of two coupled qubits that each interact with their own heat bath. Based on the collision model, we mainly study whether the decoupling approximation (i.e., the inter-system coupling is ignored when modeling the system–environment interaction) is valid or not for describing the nonequilibrium dynamics. We have first investigated the stationary heat current and steady state for symmetric systems characterized by excitation-conserving system–ancilla interactions. In this situation, we have found that the decoupling approximation is still valid to describe either heat current or steady state even at strong inter-system couplings. Then, we have turned to the asymmetric systems in the case of non-excitation-conserving system–ancilla interactions. In this case we have found that the heat current or (steady state) predicted with the decoupling approximation gradually deviates from that without this approximation as δ increases, namely, this approximation gets progressively worse with the increase of δ. In particular, without the decoupling approximation we have realized a perfect thermal rectification effect in this asymmetric systems. Through the analysis of different forms of interaction between system and environment, we have found that the thermal rectification effect is closely related to the form of system–ancilla interaction.

We have also considered the quantum discord between the two qubits without the approximation. For symmetric systems, we have found that quantum correlation between two subsystems first increases and then decreases as the inter-system coupling strength increases. The deviation of heat current (or steady state) from the decoupling approximation also increases and then decreases in this case. For asymmetric systems, the quantum correlation between two subsystems increases with the increase of inter-system coupling strength. The deviation of heat current (or steady state) from the decoupling approximation in this case increases with the increase of inter-system coupling strength. Although one would think that whether the decoupling approximation is valid or not should depend on inter-system coupling strength, we have found that it is closely related to the quantum correlations between the two subsystems. Specifically, the higher the quantum correlations means these two qubits behave more like a composite and must be treated as a whole, so the decoupling approximation gets worse, and vice versa.

## Figures and Tables

**Figure 1 entropy-23-00471-f001:**
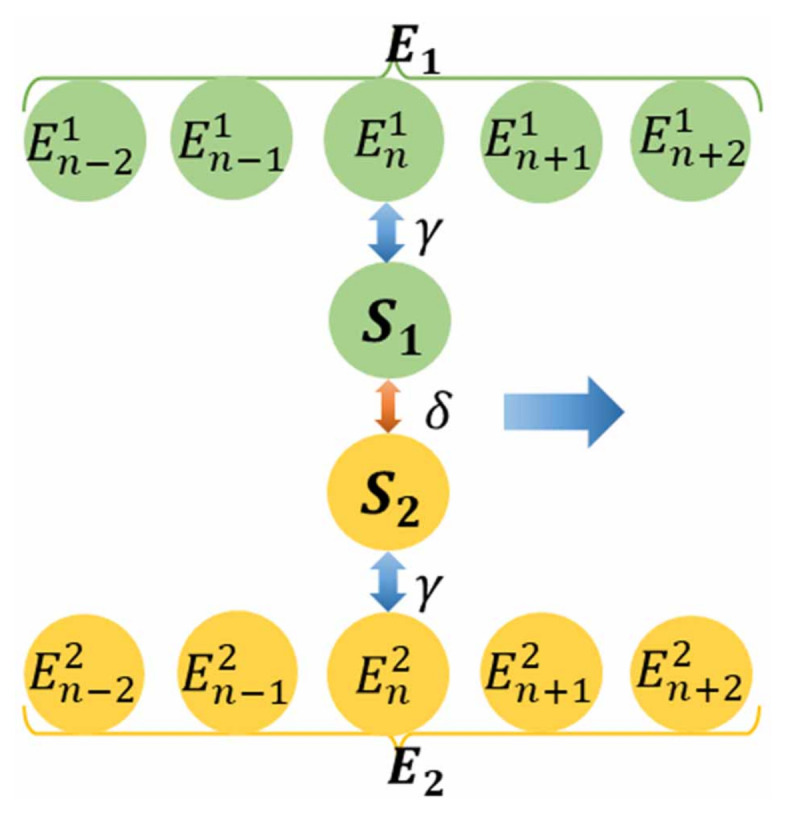
Schematic sketch of a bipartite system *S* made up of two interacting subsystems connected to two independent subenvironments. In the *n*th round of the dynamics, after a free evolution of the whole system, $S_{1}$ interacts with $E_{n}^{1}$ and next $S_{2}$ interacts with $E_{n}^{2}$. In the (n+1)th round of the dynamics, also after a free evolution of the whole system, $S_{1}$ interacts with $E_{n+1}^{1}$ and next $S_{2}$ interacts with $E_{n+1}^{2}$. The system then moves to the (n+2)th round and this process is repeated over and over.

**Figure 2 entropy-23-00471-f002:**
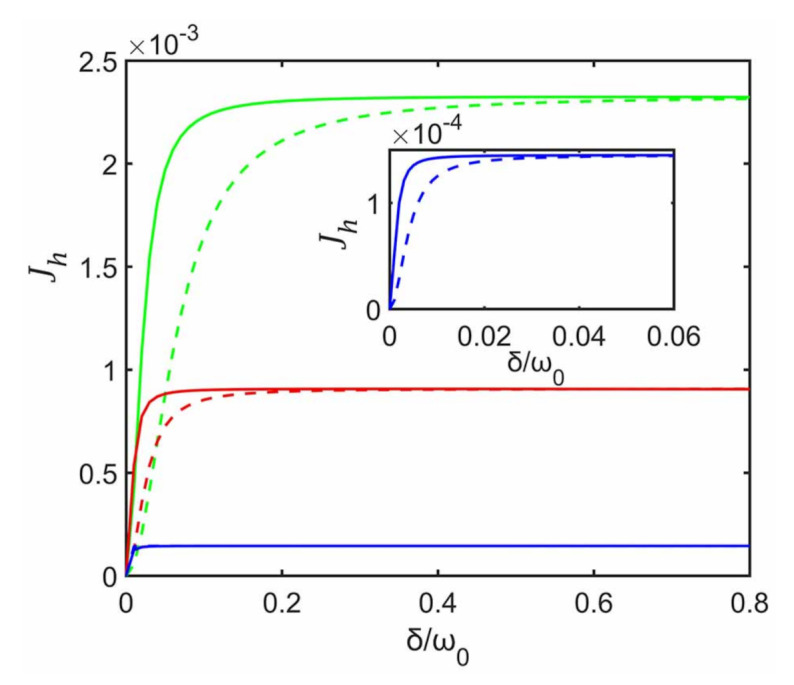
Steady heat currents $J_{h}$ as a function of δ for different γ, $\gamma =0.2\omega_{0}$ (blue), $\gamma =0.5\omega_{0}$ (red), and $\gamma =0.8\omega_{0}$ (green).The solid and dashed lines correspond to cases without and with the decoupling approximation, respectively. The inset is the magnified $J_{h}$ for $\gamma =0.2\omega_{0}$. In both cases the system is initialized in |11〉 and each ancilla is initialized in its thermal state. The plots are obtained for $\omega_{1}=\omega_{2}=\omega_{0}$, $\omega_{0}\tau =0.1$, $T_{1}=5\omega_{0}$ and $T_{2}=\omega_{0}$.

**Figure 3 entropy-23-00471-f003:**
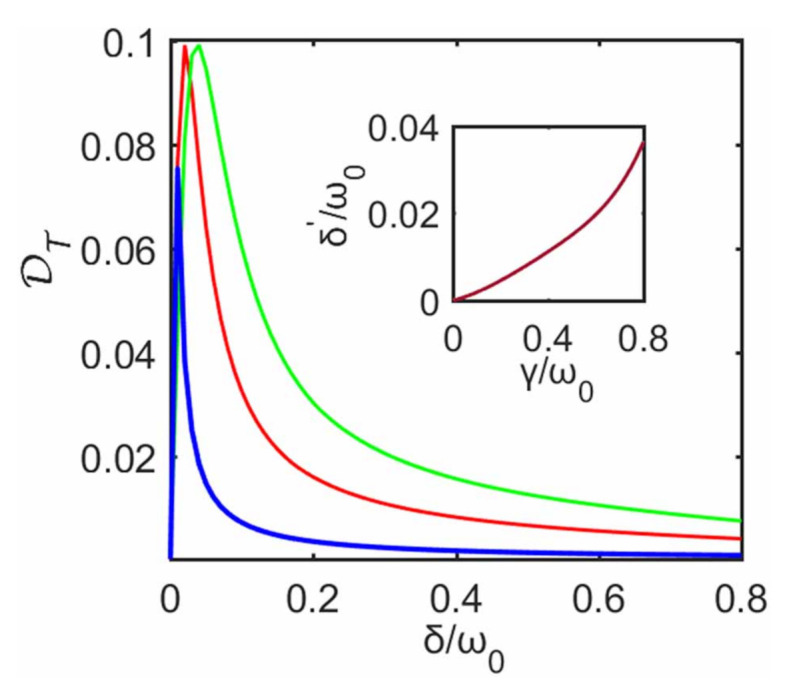
Trace distance $D_{T}$ between the density matrices obtained from cases with and without the decoupling approximation against δ at steady state. The blue line, the red line and the green line correspond to $\gamma =0.2\omega_{0}$, $\gamma =0.5\omega_{0}$ and $\gamma =0.8\omega_{0}$, respectively. The other parameters are the same as those in Figure 2. Inset shows $\delta^{\prime }$ for the maximum of $D_{T}$ as a function of γ.

**Figure 4 entropy-23-00471-f004:**
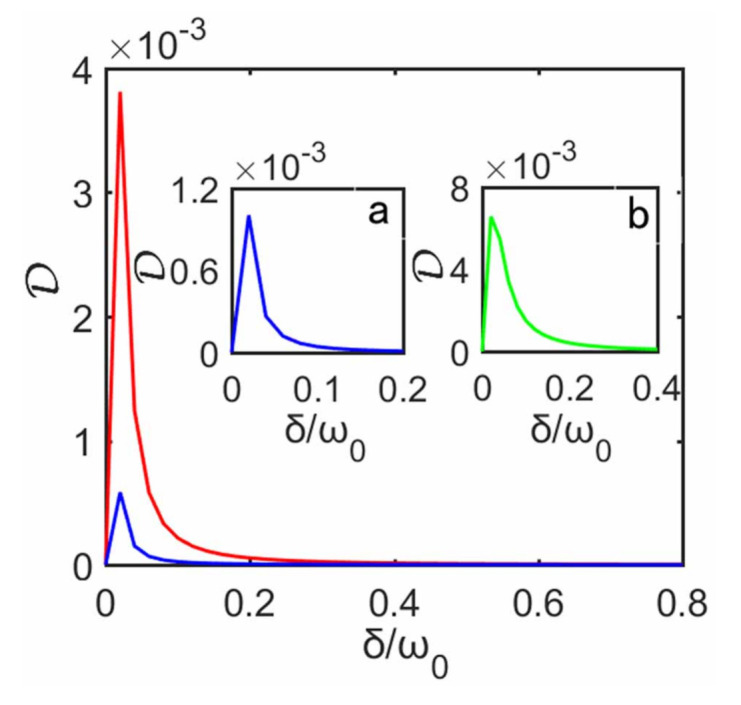
Quantum discord D of the qubit system as a function of δ, without approximation. The blue line and red line correspond to $\gamma =0.2\omega_{0}$ and $\gamma =0.5\omega_{0}$, respectively. Inset a is the magnified D for $\gamma =0.2\omega_{0}$. Inset b corresponds to $\gamma =0.8\omega_{0}$. The remaining parameters are the same as those in Figure 2.

**Figure 5 entropy-23-00471-f005:**
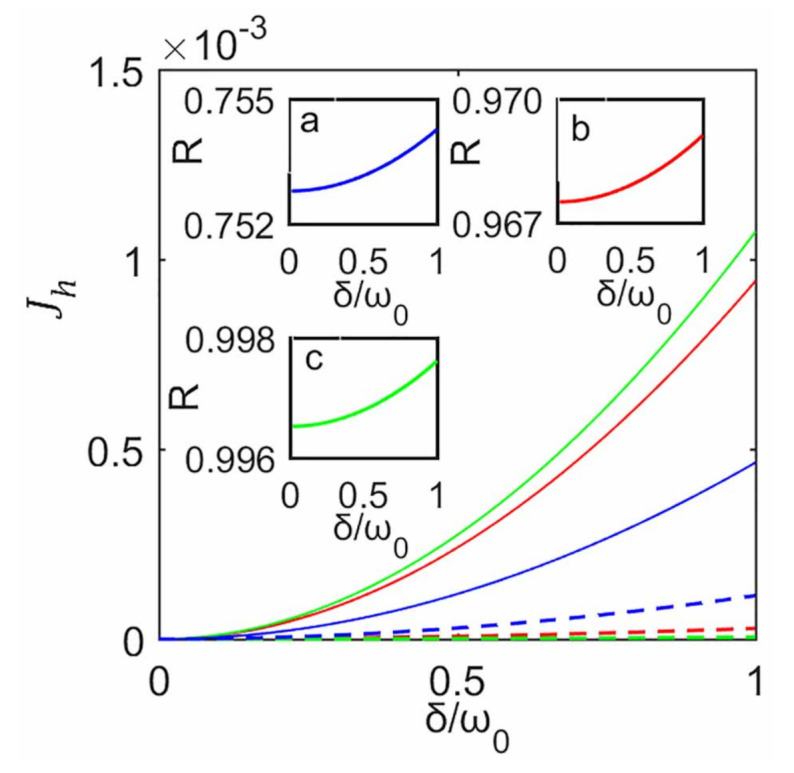
$J_{h}$ as a function of δ with various temperature difference without the decoupling approximation. We use solid line for $T_{1}=10\omega_{0}$ and different $T_{2}$, $T_{2}=0.1\omega_{0}$ (green), $4\omega_{0}$ (red), and $8\omega_{0}$ (blue); dashed line for $T_{2}=10\omega_{0}$ and different $T_{1}$, $T_{1}=0.1\omega_{0}$ (green), $4\omega_{0}$ (red), and $8\omega_{0}$ (blue). The two qubits are off-resonant with $\frac{1}{2}\omega_{1}=\omega_{2}=\omega_{0}$. Here we set $\gamma =0.3\omega_{0}$, and the other parameters are the same as those in Figure 2. Insets a, b, and c show the corresponding rectification factor *R* for $T_{1}=10\omega_{0}$ and different $T_{2}$, $T_{2}=8\omega_{0}$, $4\omega_{0}$, and $0.1\omega_{0}$, respectively.

**Figure 6 entropy-23-00471-f006:**
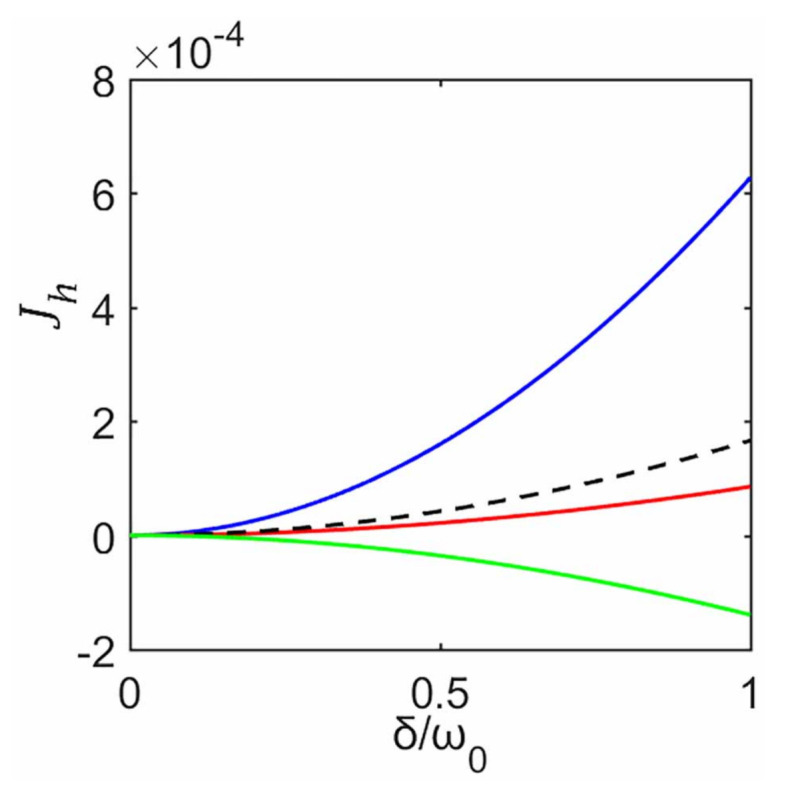
$J_{h}$ as a function of δ with various temperature difference without the decoupling approximation. The black dashed line corresponds to $T_{1}=T_{2}=10\omega_{0}$. We use blue line for $T_{1}=10\omega_{0}$ and $T_{2}=5\omega_{0}$; red and green lines for $T_{2}=10\omega_{0}$ and $T_{1}=5\omega_{0}$ and $0.05\omega_{0}$, respectively. The other parameters are the same as those in Figure 5.

**Figure 7 entropy-23-00471-f007:**
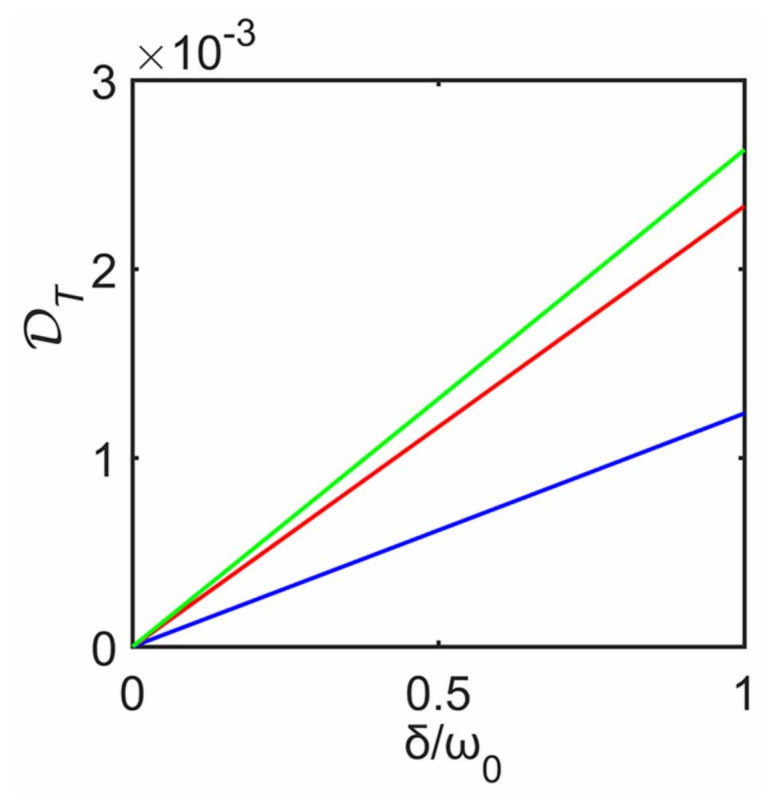
Trace distance $D_{T}$ between the density matrices obtained from cases with and without the decoupling approximation against δ for steady state, and the corresponding plot is obtained for $T_{1}=10\omega_{0}$ and different $T_{2}$, $T_{2}=0.1\omega_{0}$ (green), $4\omega_{0}$ (red), and $8\omega_{0}$ (blue).

**Figure 8 entropy-23-00471-f008:**
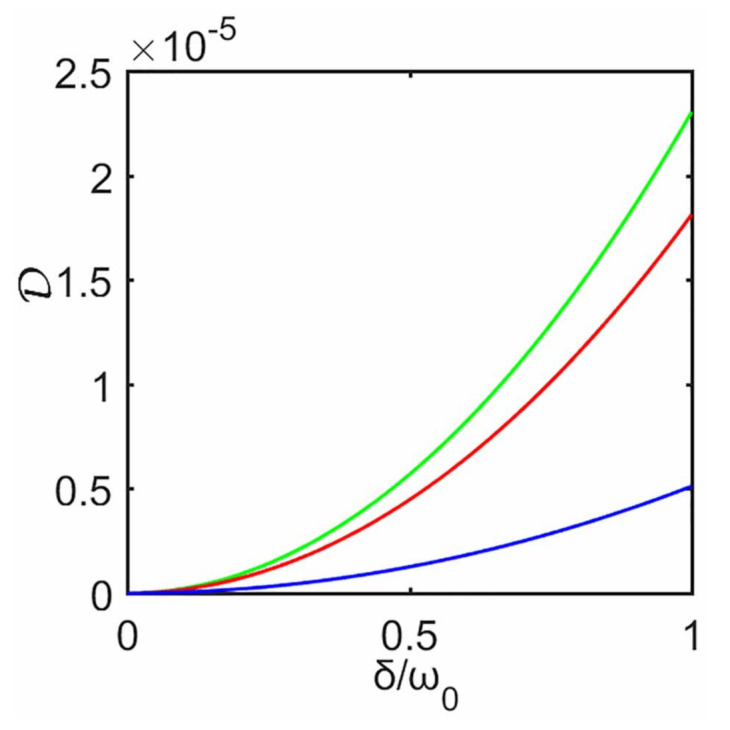
Quantum discord D of the qubit system as a function of δ, without the decoupling approximation. The plots are obtained for $T_{1}=10\omega_{0}$ and different $T_{2}$, $T_{2}=0.1\omega_{0}$ (green), $4\omega_{0}$ (red), and $8\omega_{0}$ (blue). The remaining parameters are the same as those in Figure 5.

**Figure 9 entropy-23-00471-f009:**
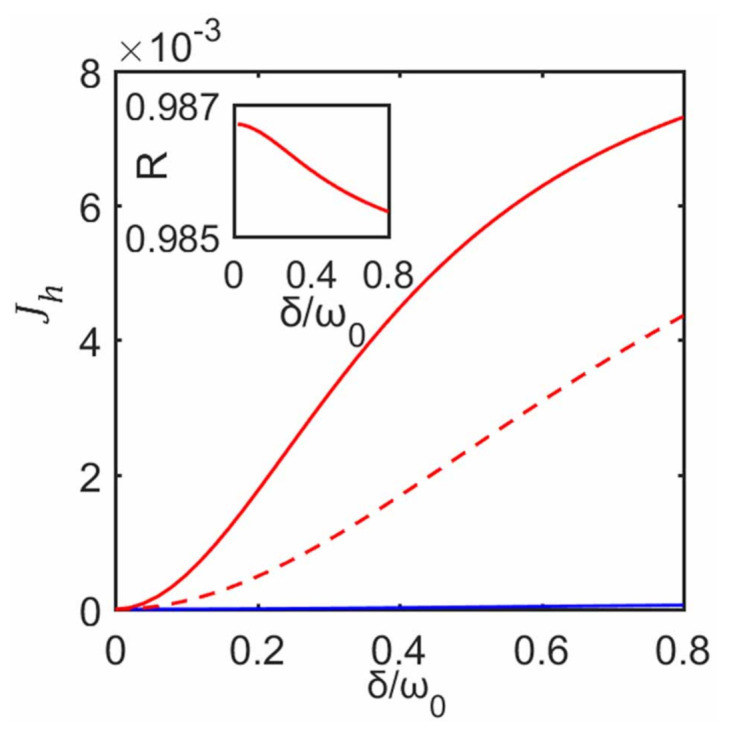
$J_{h}$ as a function of inter-system coupling strength δ. The solid and dashed lines correspond to without and with the decoupling approximation, respectively. We use red line for $T_{1}=10\omega_{0}$ and $T_{2}=\omega_{0}$. Blue line is for $T_{1}=\omega_{0}$ and $T_{2}=10\omega_{0}$ in the case without the approximation. Here we set $\gamma =0.5\omega_{0}$. The other parameters are the same as those in Figure 2. Inset shows corresponding rectification factor in the case without the decoupling approximation.

**Figure 10 entropy-23-00471-f010:**
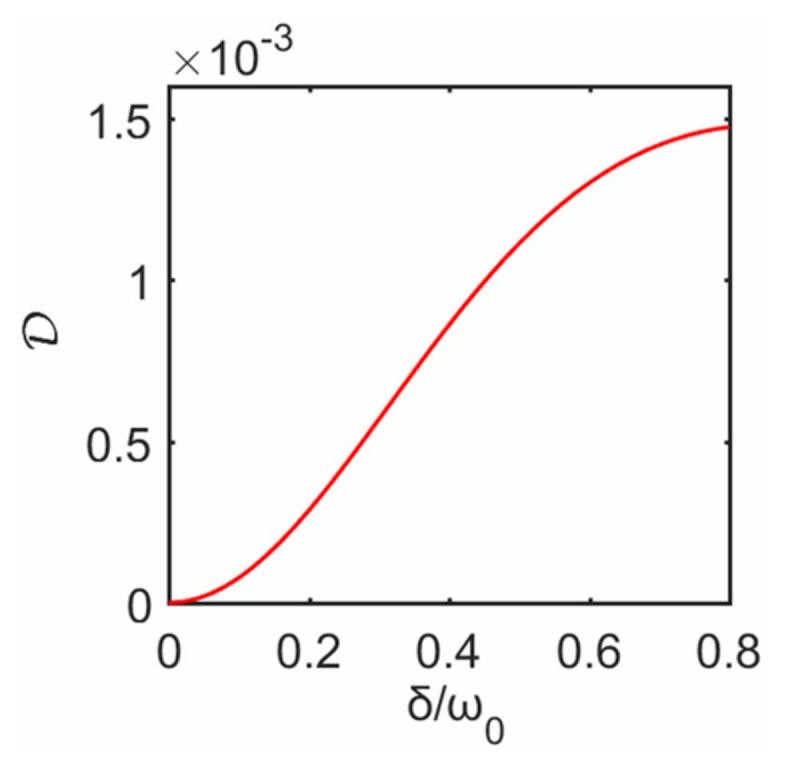
Quantum discord D of the qubit system as a function of δ, without the decoupling approximation. The plot is obtained for $T_{1}=10\omega_{0}$ and $T_{2}=\omega_{0}$. The remaining parameters are the same as those in Figure 9.

## Data Availability

Data is available from the authors.

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
