# Peer review of "Effect of Inter-System Coupling on Heat Transport in a Microscopic Collision Model"

_entropy, 2021, doi:10.3390/e23040471_

Round 1

Reviewer 1 Report

The authors study heat transport in a quantum bipartite system composed of two subsystems, each coupled to a thermal reservoir. They analyze in detail a decoupling approximation, which consists in neglecting the inter-system interaction. The authors present results showing in which cases the approximation turns out to be accurate and suggest some arguments to explain the observed phenomenology.

Even if I am not an expert of the field, the results seem to be interesting and supported by the analysis. However, the style of presentation should be improved (many typos occur, etc.). In particular, I suggest to introduce a specific name for the considered approximation (e.g. "decoupling approximation"?) in order to avoid some repetitions throughout the paper.  

In conclusion, I suggest to publish the paper, but some editing of English language is required.  

Author Response

Firstly, we would like to thank you for your affirmation. We would also like to express our sincere appreciation for your suggestion. According to your suggestion, we have rewritten our manuscript by adopting the nomination “decoupling approximation” throughout this manuscript. Besides, after carefully checking, some sentence writing has been improved, and some grammatical errors and spelling issues have been corrected in our revised manuscript.

Reviewer 2 Report

The purpose of the paper is to study the equivalent of the local approach (usually employed to simplify the derivation of the master equation of multipartite systems) when the dynamics of the open quantum system is studied using a collision model.

While the idea is appealing itself, I do not completely understand the way it is implemented in the manuscript. It seems to me that the “local” approach here simply consists of never taking into account the interaction between the qubits. Let me try to explain what I have understood about the protocol proposed, following the explanation given at the end of page 3. One has two qubits that interact to each other during some time. After that time, each of the qubits interacts with a local ancilla during the same amount of time. Then, the two local ancillas are traced out and the cycle starts again. Therefore, the two qubits interact to each other only during the time windows where there are no ancillas around. This would imply that the so-called “local” approach would just get spoiled of any qubit-qubit interaction, giving rise to two completely independent local channels, admitting obviously a local (product) steady state. This is very different from using the local approach when deriving a master equation, where the coupling term remains in the coherent part of the dynamics. Then, the task tackled by the authors is to measure the distance between the steady state of a correlated channel with respect to the local ones, which is very different from testing a local hypothesis.

The criticism I raised if of course prejudicial to the acceptance of the manuscript for publication. If my analysis is correct, the work should at least be refocused, the language should be modified, and all the results should be reinterpreted.

Other comments:

- Collision model were used in the literature to test the validity of the local master equation in a thermodynamic set-up in Physical Review Research 3, 013165 (2021).

- A general recipe for decomposing a collision model in terms of elementary processes was given in arXiv:2010.13910

- The notation is not always clear: after Eq. (9) the quantity $\hat{H}_0^\prime $ is introduced that then (right-hand-side) depends on the mute index $i$. The same happens with the “local” Hamiltonian of line 109. Besides what the authors write, the difference between Eq. (9) and Eq. (10) resides in the qubit-qubit interaction term, and not in the qubit-ancilla interaction.

- the definition of $\hat{H}_{S_i}$ and $\hat{H}_{E^n_i}$ given before Eq. (2) is not well written (there are missing parentheses).

All these things being said, I do not recommend the present form of this manuscript for publication in Entropy. However, if the work correctly is refocused and technically improved, I will be happy to change my mind.

Author Response

Thank you for your careful reading. We are very sorry for our carelessness and after carefully checking, some notation issues have been corrected in our revised manuscript (see Eq. (9) on page 4 and Eq. (10) on page 5). Besides, in order to clearly describe the difference between Eq. (9) and Eq. (10), we have rewritten the 111-114
lines on page 5.
Finally, thank you very much again for what you have done for our manuscript. And we also like to thank the referees again for their encouraging comments and suggestions which help us to improve our manuscript. If you have any requirements and suggestions, please let us know.

Round 2

Reviewer 2 Report

Following my first report, the authors improved their manuscript, which is now, in my opinion, suitable for publication in its present form